# Development of Limited-Angle Iterative Reconstruction Algorithms with Context Encoder-Based Sinogram Completion for Micro-CT Applications

**DOI:** 10.3390/s18124458

**Published:** 2018-12-16

**Authors:** Shih-Chun Jin, Chia-Jui Hsieh, Jyh-Cheng Chen, Shih-Huan Tu, Ya-Chen Chen, Tzu-Chien Hsiao, Angela Liu, Wen-Hsiang Chou, Woei-Chyn Chu, Chih-Wei Kuo

**Affiliations:** 1Department of Biomedical Imaging and Radiological Sciences, National Yang-Ming University, Taipei 11221, Taiwan; g39720010@ym.edu.tw (S.-C.J.); whchou@ym.edu.tw (W.-H.C.); 2Department of Biomedical Engineering, National Yang-Ming University, Taipei 11221, Taiwan; g39804012@ym.edu.tw; 3Institute of Biomedical Engineering, National Chiao-Tung University, Hsinchu 30010, Taiwan; s920937@gmail.com (S.-H.T.); labview@cs.nctu.edu.tw (T.-C.H.); 4Institute of Computer Science and Engineering, National Chiao-Tung University, Hsinchu 30010, Taiwan; yjchen.cs00g@g2.nctu.edu.tw; 5Department of Computer Science and Engineering, National Chiao-Tung University, Hsinchu 30010, Taiwan; 6Department of Biomedical Engineering, The University of Texas at Austin, Austin, TX 78712, USA; angela26.liu@utexas.edu; 7Materials & Electro-Optics Research Division, National Chung-Shan Institute of Science & Technology, Taoyuan 32599, Taiwan; kuocw00@gmail.com

**Keywords:** context encoder (CE), limited-angle iterative reconstruction (LAIR), generative adversarial network (GAN)

## Abstract

Limited-angle iterative reconstruction (LAIR) reduces the radiation dose required for computed tomography (CT) imaging by decreasing the range of the projection angle. We developed an image-quality-based stopping-criteria method with a flexible and innovative instrument design that, when combined with LAIR, provides the image quality of a conventional CT system. This study describes the construction of different scan acquisition protocols for micro-CT system applications. Fully-sampled Feldkamp (FDK)-reconstructed images were used as references for comparison to assess the image quality produced by these tested protocols. The insufficient portions of a sinogram were inpainted by applying a context encoder (CE), a type of generative adversarial network, to the LAIR process. The context image was passed through an encoder to identify features that were connected to the decoder using a channel-wise fully-connected layer. Our results evidence the excellent performance of this novel approach. Even when we reduce the radiation dose by 1/4, the iterative-based LAIR improved the full-width half-maximum, contrast-to-noise and signal-to-noise ratios by 20% to 40% compared to a fully-sampled FDK-based reconstruction. Our data support that this CE-based sinogram completion method enhances the efficacy and efficiency of LAIR and that would allow feasibility of limited angle reconstruction.

## 1. Introduction

Limited-angle iterative reconstruction (LAIR) is an image reconstruction method developed for to compensate incomplete data acquisition [1]. LAIR provides new acquisition flexibility and innovative instrument design that is not restricted by the size of the object [2] and also mitigates the high radiation dose required for computed tomography (CT) imaging [3]. In fact, the trade-off between image quality (IQ) and angular-sampling (related to radiation exposure) remains an ongoing issue in CT research. To date, three options exist toward managing this issue: (1) reduced X-ray exposure, (2) sparse reconstruction (extend sampling interval from a projection with full-coverage) and (3) limited-angle (LA) reconstruction (new acquisition flexibility and innovative instrument design, for example, the rotation-type tomosynthesis system). However, LA reconstruction confers insufficient angular coverage in CT acquisitions with incomplete projection data. The frequency spectrum is truncated, resulting in artifacts and difficulties in obtaining high-quality reconstructed images. This problem arises when a limited angular range is necessary for various reasons; it results from intensity inhomogeneity and streaking artifacts in the image domain.

The LAIR method is basically a sinogram recovery system for missing data in the projection domain. In our research group, we try to recover the full sinogram from an incomplete LA sinogram. Many benefits from deep learning-based algorithms have been successfully applied to the field of medical image processing, promising considerable improvements in image understanding tasks including computer-aided diagnosis [4]. Here we would like to apply a semi-supervised automatic inpainting method: context encoder (CE) [5] to recover the missing data in a sinogram. 

Based on compressive sensing (CS), the prior information involving sparse properties has attracted attention to the area of CT image reconstruction. Inspired by the theory’s success in signal processing, CS-based algorithms have been used to reconstruct images from reduced projection data [6]. The total-variation (TV) regularization method removes noise while preserving edges using an energy function [2,7]. However, this algorithm is object-dependent and no general stopping criteria are suitably applicable to every situation, especially for iterative reconstruction (IR) algorithms. Recent efforts have focused on using prior knowledge to recover signals from incomplete measurements. If the image is sparse in a domain, which has little coherence in the sampling domain, then the image can be almost entirely recovered with a few samplings whose number is proportional to the amount of non-zero entries in the sparse representation. The signal can be recovered precisely if the restricted isometry property (RIP) exists. Due to the fact that medical images are usually edge-sparse, TV minimization is often used to rectify incomplete data issues in tomography [8,9]. 

The appropriate knowledge-based method has to be retrieved from the well-reconstructed data (i.e., the training set) and used during the reconstruction of the target image (i.e., the testing set) to enable difference minimization. Dictionary learning divides images into patches to train over-complete basis and incorporates a sparse constraint into the dictionary during reconstruction [10]. The entire structure of the image as a prior during reconstruction minimizes the L2-norm-based principal component used as a penalty term, otherwise, prior geometrical information is adopted. Inspired by the fact that an image can be approximately represented by a principal components analysis of the training set, a novel feature-constrained CS algorithm is proposed for CT image reconstruction. An additional constraint is applied to the relationship between the image and the feature space during reconstruction, which is a sparse representation of the target image. Since the structural information is introduced into reconstruction, our method boosts the performance of traditional incomplete data reconstruction methods, especially for LA problems.

Applying deep learning-based method to improve IQ is a new approach for dealing with low-dose CT [11]. Some researchers proposed a residual encoder-decoder convolutional neural network (RED-CNN) for low-dose CT imaging, including vendor-specific sinogram domain filtration and iterative reconstruction algorithms. But this method is only suitable for improving the image quality with a little artifact or noisy data and it cannot be used for reconstruction of insufficient sinogram data. An approach to incorporate deep learning within an iterative image reconstruction framework to reconstruct images from severely incomplete measurement data is presented by Kelly’s group [12]. They used a convolutional neural network (CNN) as a quasi-projection operator within a least squares minimization procedure. The ill-posed inverse problems without necessarily linear forward operators is provided by Jonas’s group. These problems summaries in classical regularization theory and recent advances in deep learning to perform learning while making use of prior information about the inverse problem encoded in the forward operator, noise model and a regularizing function [13] and it is used in conjunction with the clinical cone-beam CT system by Elekta (Stockholm, Sweden). These methods involve the direct use of deep learning to map and recover projections in reconstruction mapping. However, our case does not take this approach because of different algorithm architecture we used.

Our proposed method is an integration of a smart sinogram completion method and an iterative reconstruction (IR) framework with image-quality-based stopping criteria to perform LA mode reconstruction. The stopping criteria consisted of some objective evaluation, such as peak signal-to-noise ratio (*PSNR*) and universal image quality index (*UIQI*) [14,15,16]. We also used structure similarity (*SSIM*), a *UIQI*-like index, to handle the completion of the sinogram [17,18]. Additionally, we also included the relative total-variation (*rTV*) between the reference and reconstructed images in our stopping-criteria calculations, to observe the convergence of the constraint. We hypothesize that there are significant IQ differences between the LA mode algorithm and the full-CT mode with the analytical Feldkamp (FDK) algorithm [19]. Using the proposed LAIR technique, we can significantly improve the IQ so that there are negligible differences in IQ between the LA and full-scan CT modes.

## 2. Materials and Methods

### 2.1. Context-Encoder-Based Sinogram Completion

CE is a type of generative adversarial network (GAN) architecture [5]. It is a convolutional neural network or an asymmetric auto-encoder that shares similar encoder-decoder architecture. The CE inpainting model was trained to generate the contents of an arbitrary image region conditioned by its surroundings. The context image is passed through the encoder to obtain features that are connected to the decoder using the channel-wise, fully-connected layer. The decoder then produces the missing regions in the image. In our case, we train the CE model by regressing to the ground truth content of the missing region.

In this model, the reconstruction loss (applied L2 distance of the mask region loss) is responsible for capturing the overall structure of the missing region, consistent with its context (in our case, the context information presented as the known pixels in the sinogram data).

The adversarial loss tries to make the generated prediction region look like the full-sinogram data. Thus, it has the effect of picking a particular mode from the distribution of the specific sinogram data. Here, we applied a stochastic gradient descent solver, Adam, for optimization purpose. It is an extension of the stochastic gradient descent that has recently been broadly adapted for deep learning applications in computer vision and natural language processing.

Here we chose leaky rectified linear units (ReLU) as our activation function defined by $f\left(x\right)=\{\begin{array}{c} \alpha x,\;x<0\\ x,\;x\geq 0 \end{array}$, where *α* is a small constant. The slope in the negative region can also be made into a parameter for each neuron. We normalized the activations of the previous layer of each batch and applied a transformation that maintains the mean activation close to 0 and its standard deviation close to 1.

Our CE architecture included a four-layer CNN for the encoder and a four-layer CNN for the decoder. A three-layer CNN was used as the discriminator. All of the CE model and computing architecture is presented in Table 1. The convolution kernel size for all of the architecture is 3 × 3. For the encoder and discriminator, down-sampling was applied layer-by-layer. We choose our activated function (leaky ReLU) with a slope of α=0.2. The momentum for the moving mean and the moving variance of the normalized batch is 0.8. The decoder structure is the inversed encoder and vice versa [20]. To balance the different pixel numbers between the input image and output compensated region, we transferred two fully-connected layers. In each epoch, to obtain an achievable training model, there are 32 randomly-selected sinograms for training and 16 sinograms for validation from the training sets. We train these models using three different epoch numbers (1000, 5000 and 10,000) and batch size of 32. It spent 1.4 s in training and validation per epoch. After model saving, it only needs 180 s to generate one set of completed sinograms. Finally, we also used *PSNR*, *UIQI* and *SSIM* to validate the results of the CE inpainted sinograms from the testing sets prior to LA reconstruction.

### 2.2. Derivation of Image-Quality-Based Stopping-Criteria for IR Reconstruction Algorithms

The first step of IR is the construction of a system matrix that serves as the mapping between the object and its projections on the detector. The dimensions of the detectors are *N_h_* (*h* denotes height) and *N_w_* (*w* denotes width). We assume that the forward projection process follows the linear-system model ***p*** = ***Af***, where ***f*** = <*f_i_*>, *i* = 1, 2, …, *N* is an object data vector with a size of *N* = *N_h_* × *N_w_*^2^ and ***p*** = <*p_j_*>, *j* = 1, 2, …, *M* is the projection vector with a size of *M* = *N_h_* × *N_w_*. ***A*** = *a_ij_* is the effective intersection length of the projection line *j* with pixel *i*. The general IR framework can be represented in the form of the constrained optimization problem.(1)$$f^{*}=\underset{f}{\arg\min}\left(U(f)+\lambda V(f)\right),\;s.t.\;\left|f^{*}-f\right|<\varepsilon ,\;f^{*}\geq 0$$where ***f*** is a non-negative factor, *λ* is the relaxation parameter that balances the two iterative processes between *U*(***f***) and *V*(***f***) and *ε* is the relative error. The first term *U*(***f***) is the data fidelity term, which enforces the fidelity of the ***f*** with the measured projection vector ***p***. The second term *V*(***f***), which is used to regularize the objective function, constrains the convergence of the X-ray attenuation coefficient using prior anatomical information. The final calculated ***f**** must be non-negative. In this paper, we applied two optimizers, the adaptive-steepest-descent projections onto convex sets (ASD-POCS) [8,21] and TV-constrained expectation-maximization methods (EM-TV) methods [22,23], to modify and implement in this study. The regularized term is to minimize the TV-norm, which is used for both of the IR frameworks such that(2)$${\left\|f\right\|}_{TV}=\sum_{k=1}^{N_{h}}\sum_{j=1}^{N_{w}}\sum_{i=1}^{N_{w}}\sqrt{{\left(\nabla f\right)}_{i}^{2}+{\left(\nabla f\right)}_{j}^{2}+{\left(\nabla f\right)}_{k}^{2}}$$where ∇ is the gradient operator and the notations *i*, *j* and *k* represent the unit vectors in the *x*, *y* and *z* directions. In cases where both algorithms are needed for the full-projection data to complete the sinogram, we have to apply the comparison step in iterations.

However, in the real world, we could not achieve the ideal noise-free data suitable for realistic CT scanning. Subjective evaluation is usually inconvenient, time-consuming and expensive when we consider that the accurate physical environment includes the shape of an object, the characteristics of the primary photon beams, the scatter distribution of the beams and the true attenuation from voxel to voxel. Therefore, we changed the stopping criteria from Equation (1) and applied the result from Equation (1) and the following to the objective function as follows:(3)$$\left\{ \begin{array}{l} f^{*}=\underset{f}{\arg\min}\left(U(f)+\lambda V(f)\right),\\ f^{**}=\underset{f^{*}}{\arg\max}\left(UIQI(f^{*},f)-rTV(f^{*},f)\right) \end{array}\right.,\;s.t.\;f\geq 0,f^{*}\geq 0,f^{**}\geq 0,\;PSNR(f^{**},f)>k$$where *k* = 35 dB for our experiments. The second objective function uses IQ as the stopping criteria to early stop the iterative algorithms while maintaining their IQ. This objective function is used instead of the pixel differences, $\left|f^{*}-f\right|$ in Equation (1), because IQ contains more information such as luminance, contrast, structure similarities and the shape of boundary. These properties are included in merits like *PSNR*, *UIQI* and *rTV*.

The *PSNR*-value, an IQ measurement based on the maximum possible pixel difference between the reference image ***f*** (refers to a full scan in a digital phantom experiment and reconstructed image by recovered sinogram in real-world scanning) and the iteratively estimated image ***f*****, which represents a boundary condition used to maintain sufficient IQ. In addition, [24] suggests that *PSNR* over 44 dB yields acceptable IQ, *PSNR* below 35 dB loses some signal characteristics and *PSNR*-value below 30 dB are unacceptable.(4)$$\left\{ \begin{array}{l} PSNR(f^{**},f)=10\times \log\left(\max{\left(f^{**},f\right)}^{2}/MSE(f^{**},f)\right)\\ MSE(f^{**},f)=\sum_{i=1}^{N}{\left(f^{*}-f\right)}^{2}/N \end{array}\right.$$where the notation max (***f*****, ***f***) denotes the maximum possible pixel values of the two images ***f*** and ***f***** and *MSE* (***f*****, ***f***) is the mean-square-error of the ***f*** and ***f***** pixel by pixel calculation to cover over N pixels. The *UIQI* and *rTV*-values are also calculated from the same images as well as the *PSNR*-value in Equation (3).

The *UIQI* is used to evaluate the degree of similarity between ***f*** and ***f****. The dynamic range of *UIQI* is from 0 to 1.0, where 1.0 is the ideal value, achieved when ***f*** and ***f**** are equal for all pixels, which is defined in Equation (5). (5)$$UIQI(f^{*},f)=\frac{4\mu_{f}\mu_{f^{*}}\sigma_{f^{*}f}}{\left(\mu_{f}^{2}+\mu_{f^{*}}^{2})(\sigma_{f}^{2}+\sigma_{f^{*}}^{2}\right)}$$where $\mu_{f}$ and $\mu_{f^{*}}$ denote the mean values of these images, $\sigma_{f}$ and $\sigma_{f^{*}}$ denote their standard deviations, respectively, and $\sigma_{f^{*}f}$ is the covariance of both images [17]. Another similar item is *SSIM*, which is defined in Equation (6).(6)$$SSIM(f^{*},f)=\frac{\left(2\mu_{f}\mu_{f^{*}}+c_{1})(2\sigma_{f^{*}f}+c_{2}\right)}{\left(\mu_{f}^{2}+\mu_{f^{*}}^{2}+c_{1})(\sigma_{f}^{2}+\sigma_{f^{*}}^{2}+c_{2}\right)}$$

The coefficient of $c_{1}={\left(0.01\times L\right)}^{2}$ and $c_{2}={\left(0.03\times L\right)}^{2}$ are regularization constants for the luminance, contrast and structural terms, specified as a three-element vector of non-negative real numbers, *L* is the maximum value of the dynamic range between ***f**** and ***f***. Here we followed the default setting from MATLAB. The *rTV* formula evaluates the shapes between the reconstruction results ***f***** and the reference image ***f*** and is defined as:(7)$$rTV={\left\|f^{*}\right\|}_{TV}/{\left\|f\right\|}_{TV}$$

### 2.3. Micro-CT System and Experiment Design

Figure 1 shows the laboratory-designed micro-CT system that provides 50–80 kVp characteristic X-ray beams (NS-081505 X-ray Generator, NanoRay Biotech. Co., Taipei, Taiwan) and its corresponding spectrum, which was measured using a commercial X-ray spectrometer (X-123 Completed Amptek Inc., Bedford, MA, USA) [25]. Raw projections were acquired from a flat panel detector (864 × 1536 with 75-μm pixel size; Dexela1207, Perkin Elmer Inc., Waltham, MA, USA). The distance from the source to the rotation-center (SOD) of the object is 299 mm and the source to the detector (SID) is 325 mm. The intrinsic field of view (FOV) of each projection is nearly 64.4 mm under this geometric condition. The reconstructed volume (in virtual space) FOV is 59.2 mm × 59.2 mm × 106.2 mm. In real space, the object needs to be smaller than a 25-mm radius cylindrical area to avoid critical component collision and damage during scanning. 

All developed reconstruction programs for CT run on a workstation (Precision T5600, Dell Inc., Taipei, Taiwan), which consists of two CPUs (Xeon E5-2620, 2.0 GHz, Intel Co., Taipei, Taiwan) and one GPU (Tesla C2075, NVIDIA Co., Taipei, Taiwan). The CUDA-based architecture uses parallel computing, applied on a development platform (MATLAB^®^ ver. R2017b, Terasoft Inc., Taipei, Taiwan), except the CE-based sinogram completion method. The reconstruction time using the FDK algorithm is nearly 0.3 s per slice. The iterative algorithm needs are based on the iteration number, in general, system matrix creation requires 30 s per projection. The iteration method spent an average of 300 s per slice to achieve acceptable IQ.

The CE algorithm was developed on a high-performance server in a powerful GPU (Tesla K40C, NVIDIA Co., Taipei, Taiwan) and CPU (Xeon E5-2620 v.3, 2.4 GHz, Intel Co., Taipei, Taiwan) environment. For easy implementation, we did CE processing using the Keras platform, a high-level neural network application and programming interface, written in Python and capable of running on top of TensorFlow, Computational Network Toolkit and Theano. Furthermore, we used a GPU to help speed up the training process.

### 2.4. Manipulation of the Limited Angle Sinogram

In the simulation, we used a reconstructed CT image and executed a forward projection to create a reference sinogram. The standard sampling rate is 1° within full 360° coverage. Before LAIR reconstruction, we want to recover 90° LA reconstruction by applying CE inpainting. Owing to sinogram repeating after 360°, we copied the first 90° of sinogram information to extend our sinogram to be a 450° covered sinogram as the reference sinogram. The LA sinogram was made using the reference sinogram, which was replaced by 0 during the 90–359° region.

The training set consists of four full-sinogram sets and their corresponding LA sinograms; two LA sinogram sets were used as the testing data. The details of the sinogram recovered from the CT image are described in Table 2 and illustrated in Figure 2. To expand the data variability, we were given a total of 5000 sinograms in our training data from a generated digital cylinder phantom, 2D Shepp-Logan phantom, MOBY digital mouse phantom and MINST digit numbers (from 0 to 9). The other 5000 sinograms, including 3D digital Shepp-Logan phantom, four types of quality assurance phantoms (QRM GmbH. Co., Moehrendorf, Germany) and other six mice, were served as the testing set. Table 3 describes the Monte Carlo simulation using random numbers to decide the initial image state including the direction, reflection, inversion, initial slice selection or even phantom generations from the training image used to build-up the sinograms. Table 3 also describes how we use the testing dataset for validation and evaluation of the IQ. The testing sets use data from simulated digital phantoms, physical phantoms and mice. The main concept and flowchart is illustrated in Figure 3 to describe that (1) how to generate limited-angle sinogram for training, (2) to build the CE completion model to recover sinogram data, (3) to use analytical or iterative algorithm to reconstruct the image and (4) to evaluate the recovered sinogram and reconstructed image.

### 2.5. Scanning Protocol and Experiment Design

Laboratory scanned CT data were collected from four standard quality assurance phantoms: 25-μm wire, contrast-scale, water cylinder and hydroxyl-apatite (HA) phantom (QRM GmbH, Moehrendorf, Germany) and one animal (BALB/c, Nude/SCID mouse, 19.75 g) using our micro-FT/CT molecular imaging system. There are two protocols to evaluate the two types of LAIR algorithms: the LA (1°/projection with 90° coverage) and the loosely-sampled-full-coverage (LSFC; 5°/projection with 360° coverage) modes. LSFC is used to simulate sparse sampling cases. 

### 2.6. Performance Evaluation by Physical Phantoms

For comparison, the FDK method was used in densely-sampled-full-coverage mode (DSFC; 1°/projection with 360° coverage) as a reference for the IQ assessments. The insufficient projection data needs to be recovered using the CE method before the reconstruction step. The Pearson correlation coefficient between the standard HA values and measured intensities of the five chosen ROIs from the tomographic images was determined to evaluate linearity, each of ROI is the same size as for the SNR and CNR measurements.

Figure 4 illustrates the procedure used to image the four performance phantoms to ensure our concept can be implemented on a real system. The reconstructed image from the 25-μm wire can be used to assess the spatial resolution in the radial direction of our micro-CT system. This requires two wires (one located at the center and the other at the peripheral of the phantom) to observe the spatial resolution by calculating the full-width half-maximum (FWHM) values. The contrast-scale factor allows for the evaluation of the contrast-to-noise ratios (CNRs) and we chose five 3 × 3 mm^2^ regions-of-interest (ROIs) including the upper, lower, left and right regions around the HA core and the final one at the central core to determine the CNRs. We used the same ROIs to calculate the signal-to-noise ratios (SNRs) of the water phantom. The CNR and SNR are calculated using Equations (8) and (9), respectively:(8)$$CNR=\frac{\left|\mu_{sig}-\mu_{bg}\right|}{\sqrt{\sigma_{sig}^{2}+\sigma_{bg}^{2}}}$$(9)$$SNR=\mu_{ROI}/\sigma_{ROI}$$

### 2.7. Real Animal Application and Dose Evaluation

A mouse injected with a contrast-agent (Discover ExiTron Nano-12000, nano-PET Pharma GmbH., Berlin, Germany) was scanned to validate the system’s performance in a realistic *in vivo* animal setting (a total of 80 μL was injected). The contrast-agent allows hepatic visualization for several hours after injection and helps enhance the cardiac and celiac trunk in the mouse [26,27]. We measured the radiation dose using an electrometer (CNMC Model 206 Electrometer, Best Healthcare Inc., Monroe, NY, USA) and a 0.6 c.c. Farmer chamber before the animal scan to evaluate dose reduction. The dose was calculated using the conversion factor validated by the International Nuclear Energy Council of 0.404 × 10^7^ cGy/C. The average dose was obtained by repeating the measurement method at least 10 times and we used a 20 × 20 × 35 mm^3^ poly-methyl methacrylate box to simulate the animal’s body.

## 3. Results and Discussion

### 3.1. Evaluation of Completed Sinogram by CE

Figure 5 and Figure 6 show the CE inpainted results between one of the true sinograms and the images from the testing data. All of the sinograms used for the testing and validation of CE inpainting were acquired from six reconstructed mouse cone-beam CT images. The fixed gray-level is 0 to 80 for the sinograms domain in Figure 5 and 0 to 0.2 for the reconstructed images in Figure 6. The gray-level represent the summation of the product by each voxel from CT image and their penetrated length of the specific X-ray tracing with unit of mm^−1^.

We found that the CE-based method can recover nearly the entire missing region of the sinogram (Figure 5c). From the results, we can observe not only the sinogram-domain but also the image-domain, which revealed excellent output image performance between the recovered sinogram and the true value. There are three figures-of-merit (FOMs) used to evaluate the completed sinograms and their reconstructed images: *PSNR*, *UIQI* and *SSIM*. The value for *PSNR* needs to be as high as possible. The optimal values of *UIQI* and *SSIM* are close luminance, contrast and structure [11,12].

Table 4 contains the quantitative results from Figure 5 and Figure 6 and describes an excellent image performance. In the sinogram domain, the proposed method improves the *PSNR* from 10.1623 dB to 40.2738 dB and *UIQI* and *SSIM* from 0.3095 to 0.9993, which indicates that almost all of the missing data were recovered. Next, we use the FDK and EM-TV algorithms to reconstruct the sinogram with and without applying the CE completion method.

Only applying the EM-TV algorithm without using the completed sinogram data does not work for LA reconstruction. As seen in Figure 6b,c, the images still diverge after the EM-TV algorithm. Conversely, when we apply the CE completion, the reconstructed images are similar to the reference image. When we use the FDK method, the *PSNR* increased from 13.9891 dB to 24.1986 dB, the *UIQI* value elevated from 0.5066 to 0.8865 and the *SSIM* improved from 0.8107 to 0.9552. If we apply the general form of the EM-TV method to reconstruct the completed sinogram, it generates much better results than the FDK method (*PSNR* = 25.1891 dB, *UIQI* = 0.9023 and *SSIM* = 0.9612). 

Applying our proposed IQ-based stopping-criteria method (Equation (3)) generates a slightly improved IQ (*PSNR* = 26.6432 dB, *UIQI* = 0.9106 and *SSIM* = 0.9813) compare to the original EM-TV algorithm.

### 3.2. Numerical Reconstruction Evaluation

Here we use one Shepp-Logan phantom to evaluate the performance of three algorithms. Figure 7 is the ground truth value compared with the FDK, ASD-POCS and EM-TV algorithms. Here, the IR algorithms were proposed as stopping criteria in Equation (3). Each of the following cases included the LA, DSFC and LSFC modes. Four types of artifact patterns (unsmoothed-shape edge, curved-ripple pattern, streak-artifact and radiative-pattern) were used to evaluate the IQ. In Figure 7, the limitations of the FDK algorithm are apparent. Despite its compact sampling size and computation speed, the quality of the LA reconstruction is unacceptable due to insufficient information for only 90° of projection data. In terms of the LSFC data, although the sampling coverage is sufficient, the large sampling interval amplifies some streak artifacts outside of the object. While the FDK method generates artifacts, both the ASD-POCS and EM-TV algorithms reduce the appearance of artifacts to produce acceptable images. These data were reconstructed using the IR method because they lack adequate information for reconstruction using the conventional FDK method. Although some artifacts cannot be removed entirely, these IR methods can reconstruct an image where the IQ is similar to that of the ground truth image. These results provide evidence in support of implementation in a real-world system.

### 3.3. Physical Phantom and Animal Micro-CT Image Evaluation

The substantial improvement to IQ is apparent from several FOMs, including spatial resolution, CNR, SNR and linearity (Figure 8). The proposed stopping criteria, which were similarity, signal preservation and noise reduction, were required for both the EM-TV and ASD-POCS algorithms with less than 10 iterations. The better the radial spatial resolution, the lower the FWHM value.

For the real application, we changed the geometry of the SID to 211.1 mm and the SOD to 49 mm, which changed the voxel size to 17.4088 μm. The FWHM width measurement using the FDK method is 52 μm from the central wire and 58.39 μm measured from the peripheral wire. Comparing the FWHMs of the different algorithms revealed that the proposed LAIR technique yields much better IQ than the DSFC mode using the FDK algorithm.

In both IR-reconstructed images, we improved the central spatial resolution by nearly 1.38 to 1.43-fold relative to the FDK method. The FWHM from the peripheral wire image is wider than at the center. The proposed LAIR methods provide 46.21 μm and 45.61 μm resolutions.

In terms of the other FOMs, the proposed LAIR improves the CNR by 20–30% and the SNR by 40%. Also, there is excellent linearity for all three algorithms. The correlation coefficient was 0.988 for the FDK method and almost 0.99 for both IR methods. In this article, we only discuss the contrast in high-contrast tissue because we are interested in osteoporosis applications. Most researchers are also very interested in low-density contrast to distinguish between tissue-like materials or similar-density objects. We shall address this issue in our future work with photon counting reconstruction.

The focus of this study is not the quantitative assessment of IR for a specific task; instead, it is tailored to demonstrate which type of IR algorithms can help improve IQ compared to using the traditional FDK algorithm. Using the fully-sinogram to reconstruct (DSFC and LSFC) and the LA mode, with our proposed stopping criteria and an appropriate TV regularization method, we can precisely reconstruct high-quality CT images.

Figure 8 shows that for both different slices of the same object and different objects, the proposed stopping-criteria selection is a robust method for the different IR algorithms used. While the performance on a specific imaging task is dependent on the object, the proposed method can maintain IQ without altering the parameters (e.g., image fidelity, regularization, relaxation, etc.) of each of the different IR algorithms. By using a TV-constrained data discrepancy minimization process, LAIR can efficiently and effectively reduce the artifacts introduced by LA.

The EM-TV algorithm incorporated the Poisson noise to match the realistic statistical model. We found that the FOMs of the EM-TV-based LAIR algorithm are better than those of the algebraic type. In this research, we focused on two factors for reducing artifacts in the FDK and IR methods: increasing the angular sampling interval and reducing the projection coverage.

Figure 9 displays a cross-sectional image of the contrast-injected mouse after the FDK-based reconstruction and proposed LAIRs were applied. It shows that the LAIR method represents an excellent choice for reconstructing the liver, heart, abdomen and pelvis portion without changing their shape forms and with high IQ. The results demonstrate that our IR methods are suitable for general cone-beam CT system but there are still some artifacts present (ring-artifact and beam-hardening effect) and we will resolve these in the near future (see Figure 8 and Figure 9).

This result supports us to prove that combining our proposed stopping criteria with the LAIR methods can automatically achieve optimal IQ similar to FDK reconstruction without deformation in standard mode. Figure 8 presents some remnants of fading star-shaped artifacts in the FDK reconstructed image. The IR methods can efficiently reduce most artifacts because they update the difference from iteration to iteration. However, if there is inconsistent data, the artifacts will be worse. The truncation artifact observed in Figure 9 may be caused by external-FOV data from the binary classification process; the outside-of-FOV mask (indicated by arrows) used on the initial FDK image cause data inconsistencies when performing re-projections and artifact enhancement during the forward-projection process in the IR methods.

### 3.4. Exposure Dose

Table 5 shows the average radiation dose required for each acquisition protocol. The dose was repeatedly measured at least 10 times to obtain the average dose rate, which was 0.0027 ± 0.31% mGy per second for each projection. Because the total dose using the step-and-shoot protocol is proportional to the number of projections, there are 90 projections for the LA protocol, 360 projections for DSFC and 72 projections for LSFC. The absorption doses are 0.2353 and 0.2005 mGy for the LA and LSFC modes, respectively. These doses are roughly 1/4 and 1/5 of the dose required for the DSFC mode, which is 0.9626 mGy. To summarize the dose reduction for the three different acquisition protocols by comparing the absorption dose and absorbed dose rate for the projections, the ratios are 88, 360 and 75 for the LA, DSFC and LSFC protocols, respectively. These ratios are very close to the corresponding projection number in each protocol. These data show that using either the LA or LSFC protocols can reduce the required radiation dose proportionally to the projection views.

**Table 5** Average absorption dose and dose equivalent from 10 measurements for three protocols: LA, DSFC and LSFC modes.

## 4. Limitations and Future Work

Our CE model is not applicable to variable-degree data recovery problems due to the size dependency of the architecture. This paper only serves as proof of principle to handle the 90° LA problem. An “auto-modulate” smart architecture for GAN that fits general purposes and can be widely used for other manual protocols will be tackled in our future work.

## 5. Conclusions

In conclusion, we have investigated an IQ-based stopping-criteria method, which was implemented in IR without IQ loss. Combining the proposed stopping criteria with the LAIR methods can automatically achieve optimal IQ similar to FDK reconstruction without deformation. Even though LAIRs confer many advantages, they still require the reference image to handle the convergence and over-iteration occurs when the stopping criteria is applied. Hence, we propose a CE-based sinogram completion method to solve this problem. Our data support that this method can effectively and efficiently reduce artifacts arising from insufficient projection data. This study’s findings indicate that our novel LA reconstruction method with CE-based sinogram completion can not only reduce radiation dose but also improve IQ for small animal micro-CT imaging. Overall, this approach represents a major step toward solving the issue of LA reconstruction that can potentially be applied to digital tomosynthesis.

## Figures and Tables

**Figure 1 sensors-18-04458-f001:**
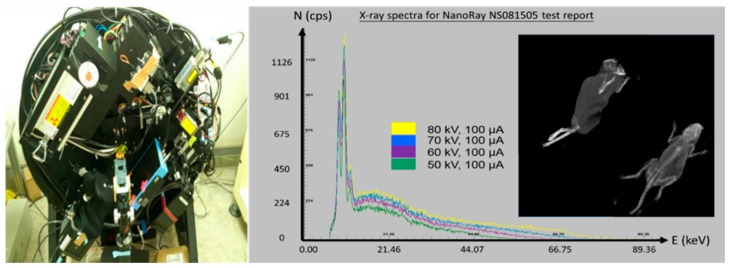
The prototype of the laboratory designed micro-computed tomography (CT) system and its 50–80 kVp X-ray spectrum.

**Figure 2 sensors-18-04458-f002:**
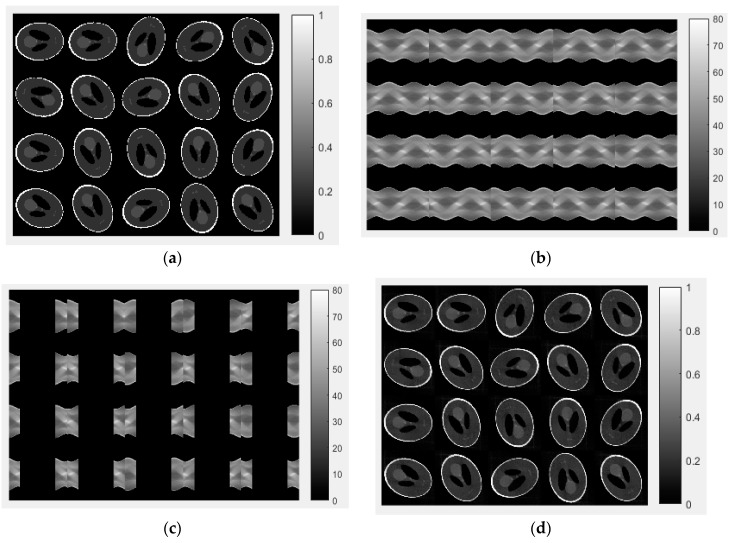
(**a**) Randomly generated 20 Shepp-Logan phantom images data using the Monte Carlo method. (**b**) Forward projection to generate corresponding sinogram data from 0 to 449°; (**c**) followed by the preprocessing rule to remove the data from 90 to 360° and to achieve the limited angle (LA) sinogram. (**d**) CT images reconstructed by context encoder (CE) inpainted sinogram.

**Figure 3 sensors-18-04458-f003:**
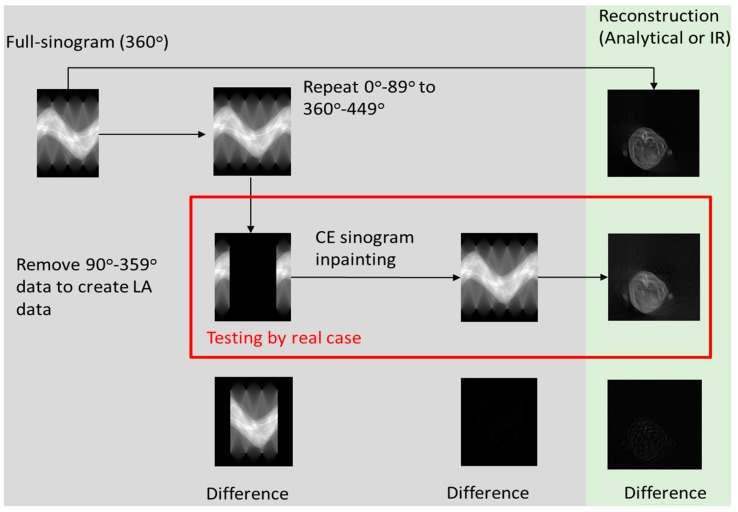
The flowchart of two-step reconstruction of the CT image by LA sinogram inpaint based method.

**Figure 4 sensors-18-04458-f004:**
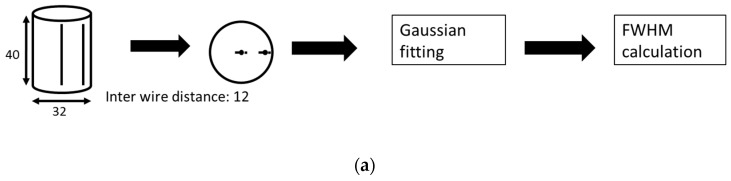
The procedure and evaluation method of the test phantom: (**a**) 25-μm wire phantom, (**b**) contrast-scale phantom, (**c**) water phantom and (**d**) Hydroxyl-apatite (HA) phantom. The dashed-block areas are the regions for calculating the contrast-to-noise ratio (CNR), signal-to-noise ratio (SNR) and linearity. The size of the phantom unit is in mm. R here means the correlation coefficient.

**Figure 5 sensors-18-04458-f005:**
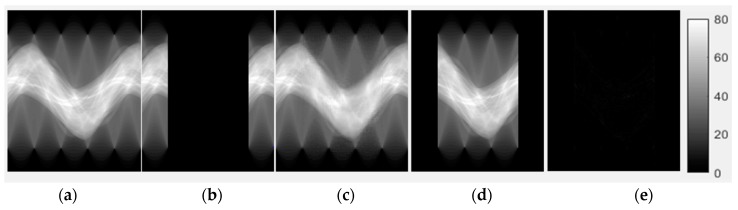
Evaluation of the completed LA sinogram results from the sinograms and their reconstructed images. (**a**) True sinogram, (**b**) Simulated LA sinogram and (**c**) CE inpainted sinogram from the testing set. (**d**) The difference of the sinogram (**a**) subtracted by (**b**) and (**e**) is the difference of the sinogram (**a**) subtracted by (**a**) and (**c**).

**Figure 6 sensors-18-04458-f006:**
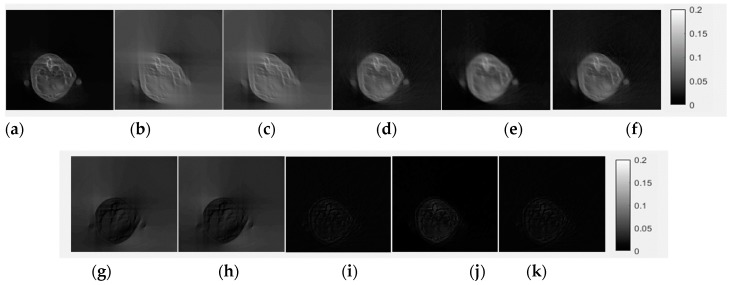
Evaluation of the reconstructed images by testing the completed LA sinograms. (**a**) The true sinogram reconstructed using the Feldkamp (FDK) algorithm (reference image). The LA sinogram without completion was directly reconstructed by (**b**) FDK and (**c**) TV-constrained expectation-maximization methods (EM-TV) algorithm (100 iterations). The CE inpainted sinogram reconstructed CT image by (**d**) FDK, (**e**) EM-TV algorithm (20 iterations) and (**f**) EM-TV algorithm + IQ-based stopping criteria (only nine iterations). The (**g**–**k**) show the difference between (**b**–**f**) compared to the reference image (**a**). The display is in the range of [0, 0.02].

**Figure 7 sensors-18-04458-f007:**
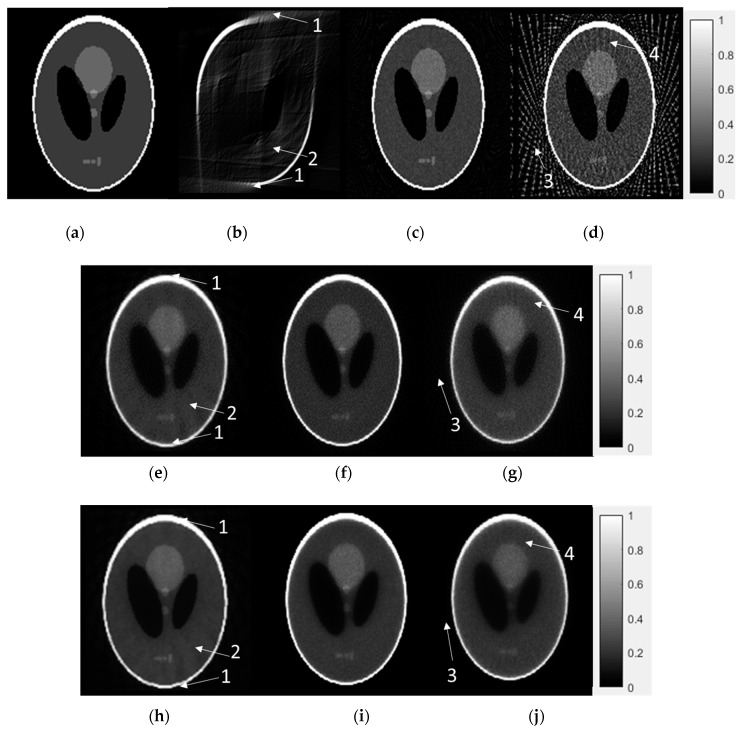
Numerical results for 3D Shepp-Logan phantom: (**a**) True image. (**b**) FDK with LA mode (1°/projection with 90° coverage), (**c**) FDK with DSFC (1°/projection with 360° coverage) mode, (**d**) FDK with LSFC (5°/projection with 360° coverage) mode. (**e**) ASD-POCS + CE inpainting + IQ-based stopping criteria for LA mode, (**f**) DSFC and (**g**) LSFC mode. (**h**) EM-TV-based algorithm + CE inpainting + IQ-based stopping criteria for LA, (**i**) DSFC and (**j**) LSFC mode. Arrows point to the four types of artifact patterns: (1) unsmoothed shape edge (2) curved-ripple pattern (3) streak-artifact and (4) radiative-pattern.

**Figure 8 sensors-18-04458-f008:**
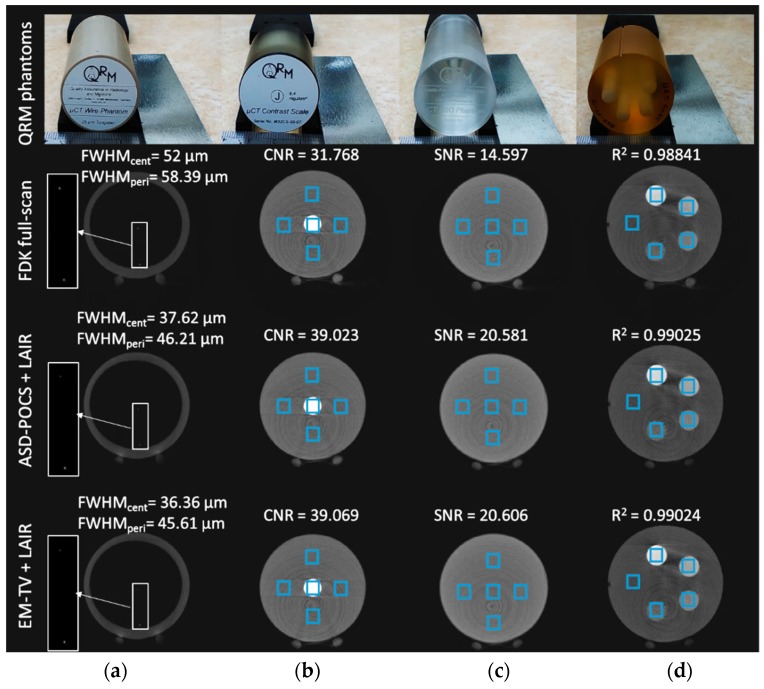
The reconstructed images from the FDK method in DSFC mode and the proposed stopping criteria included in the two LAIR algorithms (ASD-POCS and EM-TV) for four standard physical phantoms including (**a**) 25-μm wire phantom; (**b**) contrast-scale phantom; (**c**) water phantom and (**d**) HA phantom. The display’s dynamic range was set to the maximum for each different phantom image and the same scale between different algorithms automatically. For details of image analysis, please refer to Figure 4. The standard deviation of the fitted Gaussian function was calculated as 21.739 μm under the geometry of SID of 211.1 mm and SOD of 49 mm conditions to measure the radial spatial resolutions. The 3 × 3 mm^2^ ROIs were applied to the CNR, SNR and linearity measurements.

**Figure 9 sensors-18-04458-f009:**
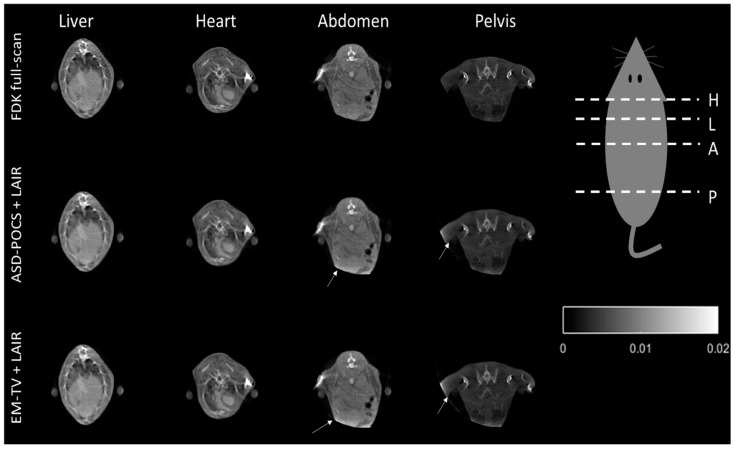
The tomographic images of the contrast-injected mouse, which cross over into the liver (L), heart (H), abdomen (A) and pelvis (P) regions, reconstructed using the conventional FDK and the two LAIR algorithms. The display is in the range of [0, 0.02].

**Table 1 sensors-18-04458-t001:** The architecture of the generative adversarial-based context encoder model on the Keras platform.

Generator		Discriminator
Encoder	Decoder	
3 × 3, d = 2, conv, ↓, LeakyRELU(0.2), BN(0.8)	34 × 17, d = 16, fully-connected	3 × 3, d = 64, conv, ↓, LeakyRELU(0.2), BN(0.8)
3 × 3, d = 4, conv, ↓, LeakyRELU(0.2), BN(0.8)	3 × 3, d = 16, conv, ↑, RELU, BN(0.8)	3 × 3, d = 128, conv, ↓, LeakyRELU(0.2), BN(0.8)
3 × 3, d = 8, conv, ↓, LeakyRELU(0.2), BN(0.8)	3 × 3, d = 8, conv, ↑, RELU, BN(0.8)	3 × 3, d = 128, conv, LeakyRELU(0.2), BN(0.8)
3 × 3, d = 16, conv, ↓, LeakyRELU(0.2), BN(0.8), Drop(0.5)	3 × 3, d = 4, conv, ↑, RELU, BN(0.8)	104,448 fully-connected
21,696 fully-connected	3 × 3, d = 2, conv, ↑, RELU, BN(0.8)	1 fully-connected, sigmoid
	3 × 3, d = 1, tanh	

LeakyRELU: leaky rectified linear units, RELU: rectified linear units, Drop: dropout layer, ↓: 2 × 2 down-sampling, ↑: 2 × 2 up-sampling, d: depth of the filtered image, BN: batched normalization.

**Table 2 sensors-18-04458-t002:** Simulation methods of image pre-processing to manipulate the LA sinogram before CE inpainting on the MATLAB platform.

Pre-Processing Procedure	Dimensions (voxel)	Voxel Size (um)	Sinogram Radial Sampling (degree)
CT reconstructed image	864 × 864 × 1536	69.00	
Slice selection (air region rejection)	864 × 864 × 500	69.00	
Binning 2 × 2	432 × 432 × 250	138.00	
ROI selection and isotropic air region rejection for training data, with flip, random rotation to do data argumentation)	380 × 380 × 250	138.00	
Forward projection to create sinogram data (replaced by 0 from the 90–359° region and repeated 0–89° information after 360–449°)	541 × 450 × 250	75.00	1.00
Resizing of the sinogram to meet the input size of the CE architecture	544 × 448 × 250	74.586	1.004
The CE inpainted sinogram	544 × 448 × 250	74.586	1.004
Reconstruction to image domain with bilinear interpolation	864 × 864 × 500	68.61	

**Table 3 sensors-18-04458-t003:** The random number generated events according to method.

Type of Object (Training)	Random Number Generated Events	Number per Slice	Total Sinograms Generated
Digital cylinder phantom	Number of cylinders (1–4), center of cylinder (initial x, y position), cylinder radius (from 10 to 20 pixel), initial phantom rotated angle (in degrees), reflection (yes/no), reversed rotation (yes/no)	250 random build per slice	1.250
2D Shepp-Logan phantom	Initial phantom rotated angle (in degree), reflection (yes/no), reversed rotation (yes/no)	250 initial random rotated per slice	1.250
MOBY digital mouse phantom	Slice number (1-208 in axial direction), initial rotated angle (in degree), reflection (yes/no), reversed rotation (yes/no)	125 slices from MOBY phantom (208 slices)	1.250
MINST dataset (Digit numbers 0–9)	Randomly select 125 images per digit	Numbers 0–9, total 10 sets	1.250
3D Shepp-Logan phantom	Center 200 from 500 slices (remove air region)	200	200
QRM quality assurance phantom	Select 450 from 500 slices, from 4 phantoms (wire phantom, contrast, water phantom and hydroxyl-appetite)	450 × 4 phantoms	1.800
Animal data	Use total 500 slice data from 6 mice (covered from head to pelvic region)	500 × 6 mice	3.000

**Table 4 sensors-18-04458-t004:** Figures-of-merit for limited angle sinogram completion in sinogram and image domain evaluation.

Domain Type	Sinogram Domain		Image Domain				
Items	LA Figure 5b	CE Figure 5c	FDK (LA) Figure 6b	EM-TV (LA) Figure 6c	FDK (CE) Figure 6d	EM-TV (CE) Figure 6e	EM-TV (CE+IQ) Figure 6f
PSNR (dB)	10.1623	40.2738	13.9891	14.0356	24.1986	25.1891	26.6432
UIQI	0.3095	0.9993	0.5066	0.5109	0.8865	0.9023	0.9106
SSIM	0.3095	0.9993	0.8107	0.8123	0.9552	0.9612	0.9813

LA: limited-angle, CE: context-encoder, FDK: Feldkamp algorithm, EM-TV: TV-constrained expectation-maximization algorithm, IQ: image-quality-based stopping-criteria method applied, PSNR: peak signal-to-noise ratio, UIQI: universal image quality index, SSIM: structure similarity.

**Table 5 sensors-18-04458-t005:** Average absorption dose and dose equivalent from 10 measurements for three protocols: LA, DSFC and LSFC modes.

Acquisition Protocol	Dose Measurement	
	Absorption Dose (mGy)	Effective Dose (μSv)
LA mode	0.2353 ± 0.30%	0.1093 ± 0.30%
DSFC	0.9626 ± 0.31%	0.4470 ± 0.31%
LSFC	0.2005 ± 0.28%	0.0931 ± 0.28%
Dose rate per projection in 1 second	0.0027 ± 0.31%	0.0012 ± 0.31%

LA: limited-angle, DSFC: densely-sampled-full-coverage, LSFC: loosely-sampled-full-coverage.

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
