# Peer review of "Development of Limited-Angle Iterative Reconstruction Algorithms with Context Encoder-Based Sinogram Completion for Micro-CT Applications"

_sensors, 2018, doi:10.3390/s18124458_

Round 1
Reviewer 1 Report
This paper proposed a deep learning method to solve the limited angle problem. Overall contents are well organized, however, following comments should be addressed.
- Introduction: More recent papers should be introduced. Using compressed sensing, low rank methods also have been used for the limited angle (view) problem. More deep learning methods should be introduced.
- In Eq (3), please do not fix the optimization with the 35 dB. If an image is very noisy, the cost function cannot reach to the 35 dB. Thus, please use “k” and authors can mention in the text: k=35 was used in our experiments.
- Could you compare results based on deeper networks (more than 4 layers)? This reviewer thinks the number of layers are too small. Can authors try to use 8 layers for encoder and decoder?
- In Fig 4, for the artifact, authors mentioned “We think that may be due to an optimization problem and we shall continue to find the evidence to deal with that.” This reviewer thinks that this may be caused by batch normalizations in testing phase. Can authors try not to use BN in testing step?
- In Fig 6, the experimental configuration with 90 degree is too ill-posed, seems impractical. How about 130~140 degree? This would be sufficient and this reviewer hopes to see center high contrast circles.
- Line 38: Remove this: “Conclusions: Indicate the main conclusions or interpretations.”
Author Response
Dear reviewer:
We will thank you for your comments.
We replied to these problems into the attached file, please check.
Wish this replied can explain the confused part of this study.
Sincerely.

Reviewer 2 Report
This paper proposed a deep network architecture (i.e., context-encoder) with an image-quality based stopping-criteria for limited-angle CT image reconstruction. The key idea is to recover the missing data in the projection domain using the generative adversarial network. The objective function of the proposed stopping-criteria is a mixture of general image quality evaluations, such as PSNR, UIQI and rTV, to achieve better visual results in experiments.
Overall I have confused feelings about this paper. On the one hand, some of the claims and models in the article are inappropriate. For example, the variable f rather than f* should be constrained in equation (1). In Line 120 in Page 3, “In case, both algorithms are needed for full-projection data to complete the sinogram …”, it is not clear to me why the equation (2) proposed in this paper doesn't require full-projection data. Another significant weakness of the paper is that the writing quality is obviously lower than the publishable standard. The descriptions on the method are not always clearly presented. There are many errors, including the wrong grammar, improper usage of words, punctuation etc.
Moreover, I think the contribution of this work is not enough. The idea of reconstructing CT image by deep networks are quite popular lately (see e.g. [1,2,3] as follows). I think this paper should review and compare with them.
[1] J. Bai, X. Dai, Q. Wu and L. Xie, "Limited-view CT Reconstruction Based on Autoencoder-like Generative Adversarial Networks with Joint Loss," 2018 40th Annual International Conference of the IEEE Engineering in Medicine and Biology Society (EMBC).
[2] Hammernik K., Würfl T., Pock T., Maier A. (2017) A Deep Learning Architecture for Limited-Angle Computed Tomography Reconstruction. In: Maier-Hein, geb. Fritzsche K., Deserno, geb. Lehmann T., Handels H., Tolxdorff T. (eds) Bildverarbeitung für die Medizin 2017.
[3] Chen, Hu, et al. "Low-dose CT with a residual encoder-decoder convolutional neural network." IEEE transactions on medical imaging 36.12 (2017): 2524-2535.
Author Response

(The authors gave the same response as above.)

Round 2
Reviewer 1 Report
Thanks for all your efforts.
Author Response
Dear reviewer:
We had edited the full manuscript to the English editor, thank you.
Reviewer 2 Report
Thank the authors for their efforts to clarify contributions and improve the English usage. Overall, for this version, I have positive impression, but still think that the presentation can be improved, especially on the description of methods. For example,
1. In lines 162 - 163, it is quite confusing for the descriptions on the stopping criteria and objective function.
2. The two sections of 2.1 and 2.2 mainly describe the proposed method, including the GAN-based inpainting, stopping criteria and objective function. It may be better to present an algorithm to describe the whole pipeline for reconstruction given sampled data, and clearly present where and how to use GAN, stopping criteria and objective function in the pipeline.
Author Response
Dear reviewer:
We have replied and followed the comments to modify the manuscript. please check our reply letter as the attached file.

Round 3
Reviewer 2 Report
I am fine with the methodologies and results, but still unsatisfactory with mathematical formulas in Eqn.(1) and (2), from the perspective of mathematics.
First, in the condition of "s.t., ..." in Eqn.(1), "|f* - f|" involves a "f" which should be a reference image for comparing to stop the optimization, in my understanding. It has same notation with the "f" in objective of Eqn.(1) which is to be optimized. Therefore, the reference image and optimized image use the same notation "f".
Second, the Eqn.(2) is also confusing, because the condition after "s.t." should be a constraint on the variables of objective function. But in the current paper, it involves another optimization problem. I understand that the condition following "s.t." is the proposed stopping criteria. My suggestion is to use text to discuss how to optimize objective function to meet the requirement, instead of formulating it as a constrained optimization problem.
Author Response
Dear reviewer:
The attached file is the reply letter. Please check.
Sincerely
